# Study of the Effect of Intake Layout on the Wavefront in a Beam Expanding System of a Telescope

**Qingpeng Zhang** [1,2,3], **Yi Tan** [1,2,3,*], **Ge Ren** [1,2,3] and **Tao Tang** [1,2,3]

[1] Institute of Optics and Electronics, Chinese Academy of Sciences, No. 1 Guangdian Road, Chengdu 610209, China; zhangqingpeng16@mails.ucas.ac.cn (Q.Z.); renge@ioe.ac.cn (G.R.); tangtao24@163.com (T.T.)

[2] Key Laboratory of Optical Engineering, Chinese Academy of Sciences, Chengdu 610209, China

[3] University of the Chinese Academy of Sciences, Beijing 100049, China

* Correspondence: tanyi@ioe.ac.cn; Tel.: +86-13153917751

**Abstract:** The main disadvantage of windowless beam expansion systems is that they cannot achieve a good sealing effect. Turbulence and impurities in the environment can easily affect the imaging and primary mirror. Thus, in this study, a matrix of small holes was introduced for inflation to form a stable and smooth flow inside the system to avoid these disadvantages. In order to study the layout of the matrix, the flow state of the model was analysed, and the Lorentz–Lorenz formula and Barron gradient operator were used for ray tracing. Simulation results show that when the matrix of small holes is arranged in 16 rows with 360 holes in each row, inflation has a lesser effect on the wavefront aberration of the system. Moreover, the root mean square (RMS) of wavefront aberration was only 0.077 μm, which was superior to the other layouts considered. Experimental results show that the RMS was 0.08 μm in this state, which is consistent with the analysis. This indicates that this analysis method can meet actual work needs. The calculation methods and calculation results have high reliability and, thus, can be also used in similar situations.

**Keywords:** beam expanding system; intake system; numerical simulation; wavefront aberration; ray tracing

## 1. Introduction

The emergence and development of optical telescope systems have provided effective tools for helping people to understand the universe. In addition, optical telescopes also play an increasingly important role in target ranging, astronomical observation, earth science, laser communication, and national production. With the development of science and technology, higher requirements have been placed on the spatial resolution of these telescopes and their ability to gather light. According to the Rayleigh criterion, the most effective way to improve the spatial resolution of a telescope system is to increase its aperture. Moreover, increasing the aperture is also an important means of increasing the light collection capacity of a telescope system [1,2]. Due to limitations on the fabrication and processing of optical crystal windows, however, it is difficult to obtain suitable optical windows for large-aperture optical systems [3]. Instead, an active intake approach can be used to prevent dust, water vapour, and other impurities in the external environment from entering a windowless beam expanding system and damaging the primary mirror. In this approach, a stable, undisturbed airflow is continuously introduced into the beam expanding system to replace its current internal gas, thus forming a stable flow field inside the system [4–6]. However, with any change in the density of the flow field, the original path of a light beam propagating through the system is also changed, which will result in deflection or a phase change of the light, and thus cause image offset, blur, jitter, energy loss, and other imaging errors on the imaging plane; these phenomena are called aero-optical effects. A large number of studies

point out that the aero-optical effect of the flow field is essentially caused by the instability of the refractive index field. Reducing the instability of the refractive index field of the flow field is the most fundamental way to reduce the aero-optical effect [7–9].

Many scholars have reported significant research conclusions related to aero-optical effects. For example, Ryan Danks et al. and Myung K. Cho et al. studied the impact of external environmental loads on the imaging capabilities of telescope systems [10,11]. Adam E. Smith et al. conducted relevant research on aero-optics [12,13]. Chen Wentao et al. studied the influence of the flow field in the internal channel on optical system performance [14]. However, most scholars focus on external fluidity, the internal channel and the flow field control. The effect of intake layout on the wavefront in beam expanding systems is still unclear.

In this paper, we focus on the effect of intake layout on the wavefront in a beam expanding system. A gas intake system is proposed for the beam expanding system of the 850 mm optical telescope. In this system, the matrix of the intake is used to inflate the beam expanding system to obtain a relatively stable internal flow. To be specific, we design a model of the intake, in which the number of rows and the number of air inlet vents per row are assumed for two variables which we called variable I and variable II. We hope to find the relative optimal value of the two variables through simulation under certain conditions, so as to minimize the influence of the flow field on the beam in the expanding system and minimize the wavefront error of the beam passing through the beam expanding system. At the same time, it is necessary to find an analysis method suitable for this kind of work. In short, we hope to reduce the influence of the flow field on the light beam through the system by changing the inlet layout.

## 2. The Method of Ray Tracing

To realize ray tracing in the flow field, we must convert the flow field to the refractive index field, and then the light field data can be obtained through ray tracing in the non-uniform refractive index medium [15–17]. Thus, there is a need to understand the following theory.

### 2.1. Effect of a Flow Field on Light Transmission

The optical refractive index of a gas is affected by several state parameters of the gas [15]. This article assumes that the refractive index of a gas is determined by the gas density under the condition of room temperature. For a discrete spatial density distribution, the refractive index can be calculated by corresponding to any point using the Lorentz–Lorenz formula [15,18]:

$$n = 1 + K_{GD} \cdot \rho, \tag{1}$$

where $n$ and $\rho$ denote the index of refraction and the density respectively, and $K_{GD}$ is the Gladstone–Dale constant, which is weakly dependent on the optical wavelength. The wavelength dependence is given by the following equation [2]:

$$K_{GD} = 2.23 \times 10^{-4} (1 + \frac{7.52 \times 10^{-3}}{\lambda^2}), \tag{2}$$

where $\lambda$ denotes the wavelength and the unit of $K_{GD}$ is $m^3/kg$ .

Using the refractive index calculated via Equation (1), the optical path length (OPL) of a beam passing through the beam expanding system can be calculated accurately using the following formula:

$$OPL(x,y) = \int_0^L n(x,y,z)dz. \tag{3}$$

Due to the instability of the flow field, there will be a certain density gradient in the flow field in the beam expanding system, which will result in differences in the refractive index along the

beam propagation path. These differences will lead to deflection, jitter, and phase changes of the beam, and result in a difference relative to the ideal optical path length, called here the optical path difference (OPD):

$$OPD = OPL(x, z) - < OPL(x, z) >. \tag{4}$$

*2.2. Ray Tracing in the Non-Uniform Refractive Index Medium*

In the process of ray tracing, we use Equation (1) to map the discrete density field data from the fluid simulation to a refractive index field. Since the refractive index field is not regular, an exact analytical solution to Equation (1) cannot be obtained. Thus, based on the concept of differentiation, we divide the beam into small segments along the propagation path for the ray tracing process. We assume that within each segment, the refractive index of the medium remains approximately unchanged. The refractive index within each microelement can be approximated as the index at the centre point of that microelement. Thus, the following formula can be obtained:

$$OPL = \int_0^L n(s)ds = \sum_{i=1}^{N} OPL_i = \sum_{i=1}^{N} l \times n_i, \tag{5}$$

where $OPL_i$, $l$, $n(s)$, and $n_i$ denote the optical path length, geometric length, refractive index profile, and mean index of refraction of the $i$-th microelement respectively, and $N$ is the total number of microelements. According to the differential hypothesis, a refractive index gradient exists at both the incident and exit surfaces of each microelement. Therefore, according to Snell's law:

$$n_1 \sin \theta_1 = n_1 \sin \theta_2, \tag{6}$$

When a beam passes through each interface of the element body, its propagation direction will be deflected. Since the isosurface of the refractive index field in the beam expanding system is not a plane but an irregular surface and the incident plane of a microelement cannot be either coincident with or parallel to a constant refractive index surface, we cannot simply take the surface normal of each element body as the normal to the incident plane for a ray. Therefore, we use the Barron gradient operator to solve for the gradient of the refractive index [19,20]:

$$\begin{aligned} \frac{\partial n(i,j,k)}{\partial x} &= \frac{1}{12}n(i-2,j,k) - \frac{8}{12}(i-1,j,k) + \frac{8}{12}n(i+1,j,k) - \frac{1}{12}n(i+2,j,k) \\ \frac{\partial n(i,j,k)}{\partial y} &= \frac{1}{12}n(i,j-2,k) - \frac{8}{12}(i,j-1,k) + \frac{8}{12}n(i,j+1,k) - \frac{1}{12}n(i,j+2,k) \\ \frac{\partial n(i,j,k)}{\partial z} &= \frac{1}{12}n(i,j,k-2) - \frac{8}{12}(i,j,k-1) + \frac{8}{12}n(i,j,k+1) - \frac{1}{12}n(i,j,k+2) \end{aligned} \tag{7}$$

where $\left( \frac{\partial n(i,j,k)}{\partial x}, \frac{\partial n(i,j,k)}{\partial y}, \frac{\partial n(i,j,k)}{\partial z} \right)$ is the direction of the refractive index gradient at the point $(i, j, k)$. In this way, we can calculate the phase change and deflection of the light in the beam expanding system using the simultaneous Equations (5), (6) and (7).

The beam quality of a beam expanding system can be evaluated by calculating the wavefront aberration of a beam passing through the system. Moreover, Zernike annular polynomials can be used to accurately fit the annular wavefront aberration to effectively extract the components of the wavefront aberration, which can serve as an important basis for wavefront aberration correction. Therefore, in this paper, the wavefront aberration obtained through simulation is fitted with Zernike annular polynomials, and the compositions of the OPD profiles under different inlet conditions are compared to determine the optimal intake layout [21,22].

## 3. Simulation Analysis

### 3.1. Intake Layout of the Beam Expanding System

To verify the influence of the flow field on the beam quality in a beam expanding system, a typical Cassegrain reflection system with a diameter of 850 mm was considered as the research object. The structure is shown in Figure 1. The dissipation velocity of turbulence can be greatly increased by directly stimulating the turbulence at a small scale [23–25], so the large-scale intake holes are replaced by a matrix of small-scale holes to increase the dissipation velocity in this work. In order to determine the ideal intake layout, we designed the simulation process shown in Figure 2. According to the simulation process, the layout of the intake holes should be designed first. Computational fluid dynamics (CFD) was used to simulate the flow field to obtain the flow field distribution. After obtaining the flow field distribution, the ray tracing method mentioned in Section 2.2 was used to perform ray tracing in the non-uniform refractive index medium. Finally, the flow field distribution and wavefront data were used as the references to improve the structure to obtain a better intake layout. Through a preliminary simulation analysis, it was found that when the mass flow rate and opening rate are fixed, the flow field in the beam expanding system improves significantly with an increase in the number of rows of air inlet holes and in the number of air inlets per row. For an air intake system with one row consisting of 18 holes, the variation of root mean square (RMS) of the wavefront aberration of a beam passing through the beam expanding system is 0.32 nm. When the number of air inlet vents is increased to 36, the RMS value of a beam passing through the system is 0.26 nm. When the air intake layout is expanded from a single row of holes to a double row with 18 holes in each row, the RMS value is 0.25 nm. With an increase in the number of air inlets per row and in the number of rows in the intake system, the influence of the flow field in the beam expanding system on the beam quality gradually weakens. Therefore, we preliminarily designed intake system layouts consisting of 4, 8, and 16 rows with 90 holes per row (see Table 1) and carried out corresponding numerical simulations. Since the existing equipment can provide a flow of only 100 g of gas per second, the mass flow rate was controlled to be 100 g per second during the processes of design, simulation, and experiment.

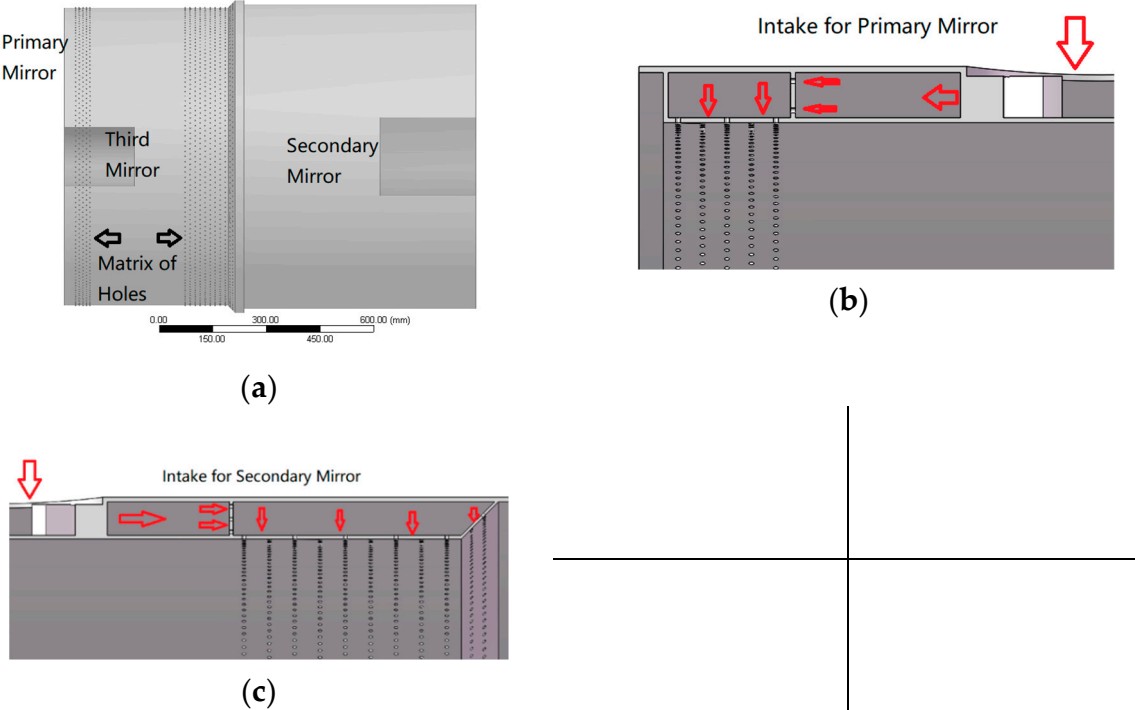

**Figure 1.** Schematic diagram of the beam expanding system: (**a**) the beam expanding system; (**b**) intake for primary mirror; (**c**) intake for second mirror.

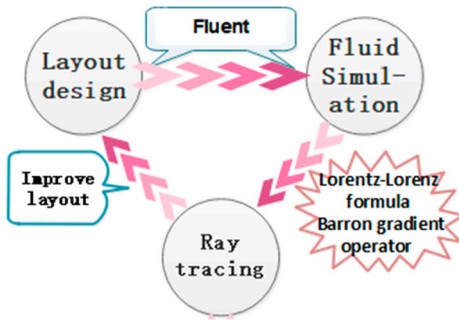

**Figure 2.** Simulation model of the beam expanding system.

**Table 1.** Initial simulation schemes.

| Project | Number of Rows | Number of Holes Per Row |
|---------|----------------|-------------------------|
| Plan A | 4 | 90 |
| Plan B | 8 | 90 |
| Plan C | 16 | 90 |

### 3.2. Simulation Analysis: Effect of Flow Field on Wavefront

In accordance with the design introduced above, we gridded the flow field in the beam expanding system; the result is shown in Figure 3.

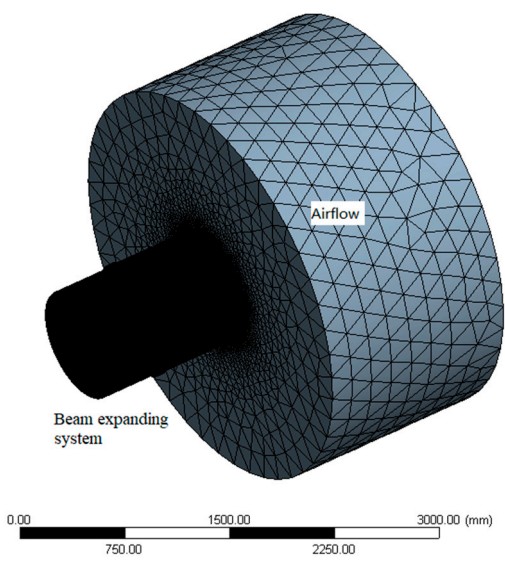

**Figure 3.** Simulation model of the beam expanding system.

In such a simulation, the airflow is mainly used to observe the spread of airflow expelled after the beam expanding system. For the flow field simulation, we used Fluent (CFD calculation software) for calculations. For simulation purposes, we set the calculation parameters as shown in Table 2.

**Table 2.** Settings of the computational fluid dynamics (CFD) software.

| Item | Value |
|------|-------|
| Material | Ideal gas |
| Model | Realisable $k - \varepsilon$ |
| Inlet | Mass flow rate |
| Mass flow rate | 0.1 kg/s |

**Table 2.** *Cont.*

| Item | Value |
|---|---|
| Supersonic/initial gauge pressure | 101,325 Pa |
| Outlet | Pressure outlet |
| Operating conditions | 0 Pa |

Because the state of the flow field is low-speed and single-phase flow, the pressure-based solver was selected. All the surfaces of the airflow region were set as outlet, and the body tube was set as the wall. Since the medium is an ideal gas, gravity was ignored during this simulation. We considered the operation of the beam expanding system after the flow field reached stability so that we could calculate the steady flow field directly. The calculated distributions of the flow field in the beam expanding system are visualized in Figures 4 and 5.

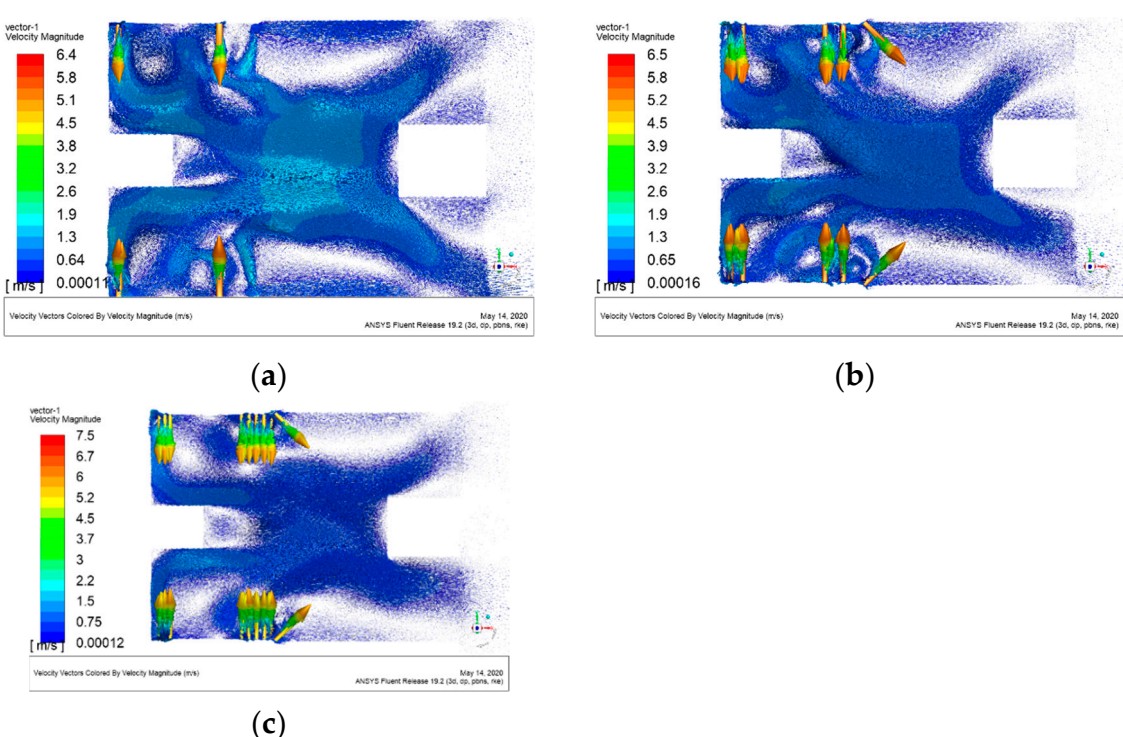

**Figure 4.** Velocity vector diagrams of the flow field in the beam expanding system: (**a**) velocity vector diagram for plan A; (**b**) velocity vector diagram for plan B; (**c**) velocity vector diagram for plan C.

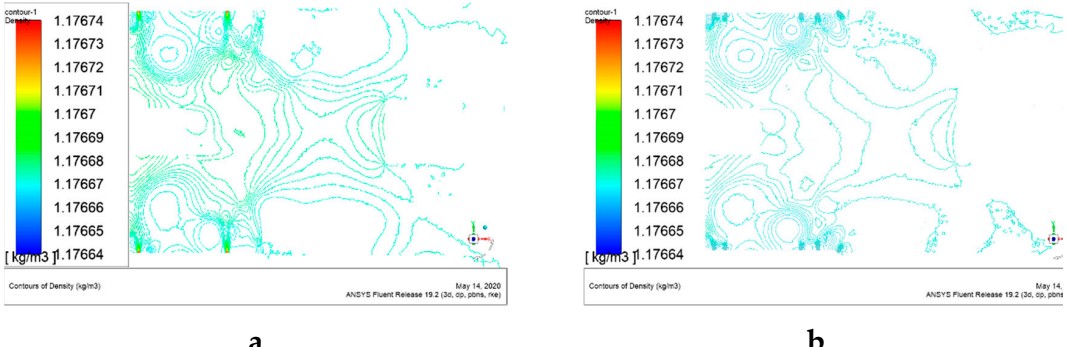

**Figure 5.** *Cont.*

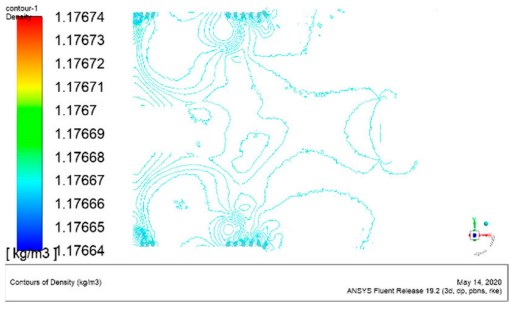

**c**

**Figure 5.** Contour maps of the fluid density in the beam expanding system: (**a**) contour map for plan A; (**b**) contour map for plan B; (**c**) contour map for plan C.

As shown in these figures, the flow field in the beam expanding system gradually improves as the number of inlet rows increases. When there are only four rows of air holes, there are two obvious large eddies in front of the main mirror, and there are several small eddies near the air intakes and in front of the third mirror. The directions of the velocity vectors at the inlet holes are obvious. As seen from the density contour map, the density contour lines in the beam expanding system are dense with multiple centres. This indicates that the flow field is rather disordered in this state. The fluid density gradient is large, and the gradient directions are disorderly. When the number of air inlet rows is increased to eight, several small eddies around the air intakes merge into several large eddies. At the same time, the directions of the velocity vectors at the inlet holes are weakened. From the density contour map, it is found that the density gradient decreases and the density contours become somewhat sparser. This indicates that the flow field in the beam expanding system gradually improves with an increase in the number of air inlet rows. When the number of air inlet rows is increased to 16, the two large vortices in front of the primary mirror also become less obvious. The several small eddies near the air intakes are combined into two larger eddies. From the density contour map, it can be seen that the contour lines are sparse; the number of density contour lines is higher only near the air intakes. This indicates that the flow field in the beam expanding system is generally of good quality; only the flow field around the inlet holes is poor.

Based on the above analysis, we designed two additional simulation scenarios, as shown in Table 3.

**Table 3.** Additional simulation schemes.

| Project | Number of Rows | Number of Holes Per Row |
|---------|----------------|-------------------------|
| Plan D  | 16             | 180                     |
| Plan E  | 16             | 360                     |

The corresponding simulation results are presented in Figures 5 and 6.

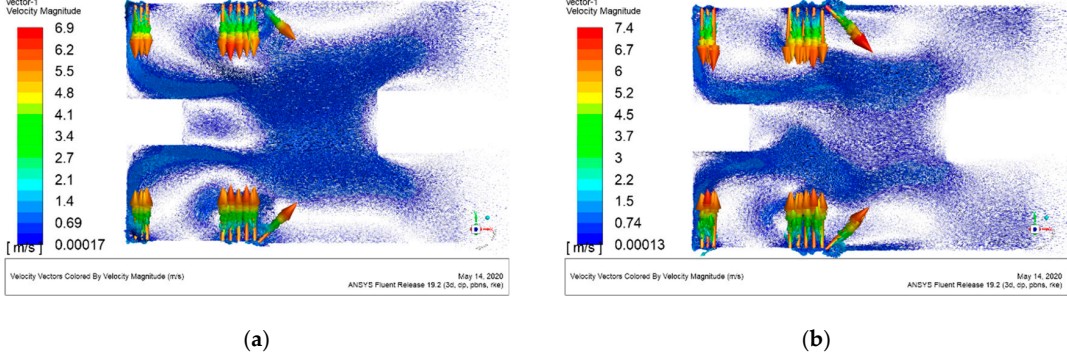

(**a**)  (**b**)

**Figure 6.** Velocity vector diagrams of the flow field in the beam expanding system for plans D and E: (**a**) velocity vector diagram for plan D; (**b**) velocity vector diagram for plan E.

By observing the velocity vector diagram for plan D, it is found that the weak eddy currents near the inlet holes are significantly reduced compared with plan C. The eddies in front of the third mirror change from two obvious eddies to a single weak eddy. The density contour lines are obviously sparser, indicating that the flow field in the beam expanding system tends to be simpler and the density gradient is decreased (Figure 7). When the number of holes per row is increased to 360, the flow field vortex in front of the third mirror essentially disappears, and those eddies in front of the main mirror are significantly weakened. The density contour lines of the flow field tend to be simple and sparse, indicating that the flow field in the beam expanding system is simple and that the density field distribution is highly uniform.

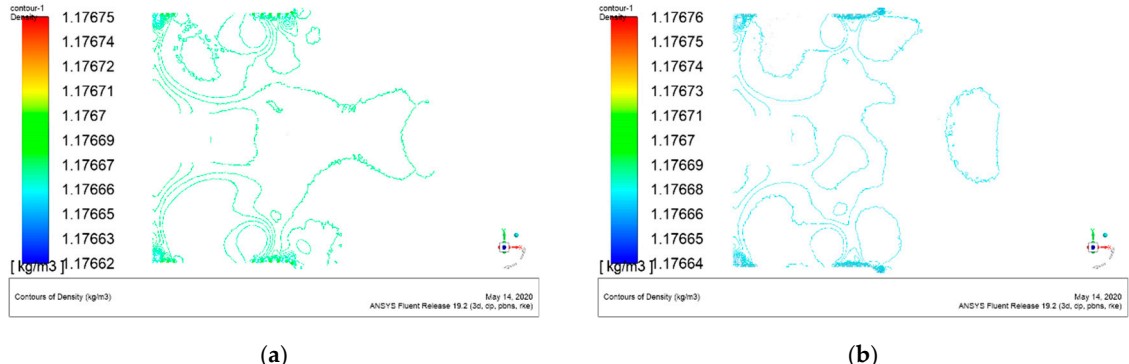

(**a**)　　　　　　　　　　　　　　　　　　　　　　　　　　(**b**)

**Figure 7.** Contour maps of the fluid density in the beam expanding system for plans D and E: (**a**) contour map for plan D; (**b**) contour map for plan E.

Below, we analyse the last three plans from the perspective of the wavefront aberration. Following the ray tracing process described in Section 2.2 above, we conducted ray tracing analyses of the optical paths in the beam expanding system. The PV (peak-to-valley) and RMS of wavefront aberration under these three plans are listed in Table 4. By comparing the wavefront aberrations of a beam passing through the beam expanding system, we can clearly see that plan D is significantly superior to plan C. This finding indicates that when the number of holes per row is 180, the flow state in the beam expanding system is better than it is with 90 holes per row. By comparing plan D with plan E, we can see that when the number of holes in a single row is further increased, and the further improvement in the wavefront aberration is significantly weaker. Thus, an increase in the number of holes per row from 180 to 360 has little effect on the wavefront aberration. However, from Figure 8, we can clearly see that the 4th-order Zernike coefficient under plan E is increased compared with that under plan D, whereas the 22nd-order coefficient is significantly decreased. These findings indicate that when the number of holes per row is increased from 180 to 360, the low-frequency components of the wavefront aberration increase in strength, while the high-frequency components are reduced, which is helpful for facilitating other methods of correction to improve the quality of the beam.

**Table 4.** Peak-to-valley (PV) and root mean square (RMS) of wavefront aberration under different schemes.

| Project | PV | RMS |
|---------|-------|-------|
| Plan C | 0.214 | 0.049 |
| Plan D | 0.152 | 0.032 |
| Plan E | 0.144 | 0.031 |

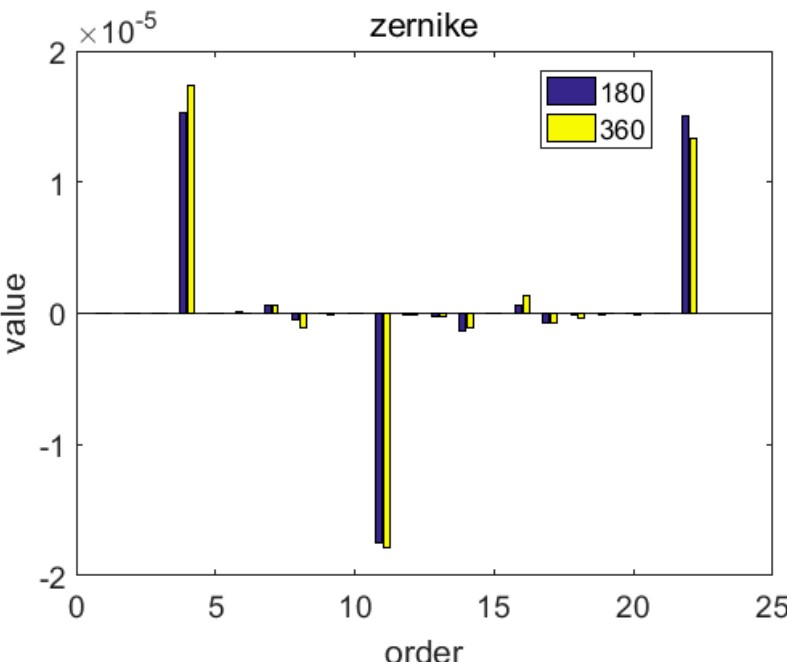

**Figure 8.** Zernike coefficients of the wavefront aberrations for plans D and E.

Through the above analysis, we found that when the number of air inlet holes in a single row is 360, the beam passing through the flow field inside the beam expanding system has the best quality. Therefore, when designing our experiment, we used plan E as the experimental scheme to verify the influence of the intake system on the flow field in the beam expanding system and on the light passing through the system.

## 4. Experimental Verification

To verify the accuracy of the simulation process above and the credibility of the simulation results, we designed the following experiment. The experimental platform is shown in Figure 9. A beam of parallel light is emitted by the collimator (Figure 9a), passes through the beam expanding system (Figure 9d), and then reaches the Shack–Hartmann wavefront sensor (Figure 9c). The gas being introduced into the beam expanding system is provided by the air supply system (Figure 9f), which is connected to gas cylinders (Figure 9e). For the distribution of the inlet holes, we adopted the optimal plan E introduced above; the corresponding layout is shown in Figure 9. The total number of inlet holes is 5760, divided into 16 rows with 360 holes in each row (Figure 10). During the intake process, the temperature and inlet pressure of gas for the beam expanding system are controlled by means of the intake control system to avoid large errors due to the changes in the refractive index that would be caused by changes in the temperature and pressure of the filling gas. During the experiment, we set the intake time to 50 s, and the wavefront aberration measurements were collected by the wavefront sensor (Figure 11). Figure 11a shows that before the start of gas intake, the peak-to-valley (PV) and RMS values of the wavefront aberration were small and flat. When we began injection at 18 s, the wavefront aberration increased rapidly; the RMS value approached 0.62 µm before levelling off. Subsequently, when the gas flow was stopped at 68 s, the RMS wavefront aberration rose to nearly 1 µm and then quickly dropped to the pre-flow level, levelling off in approximately 100 s.

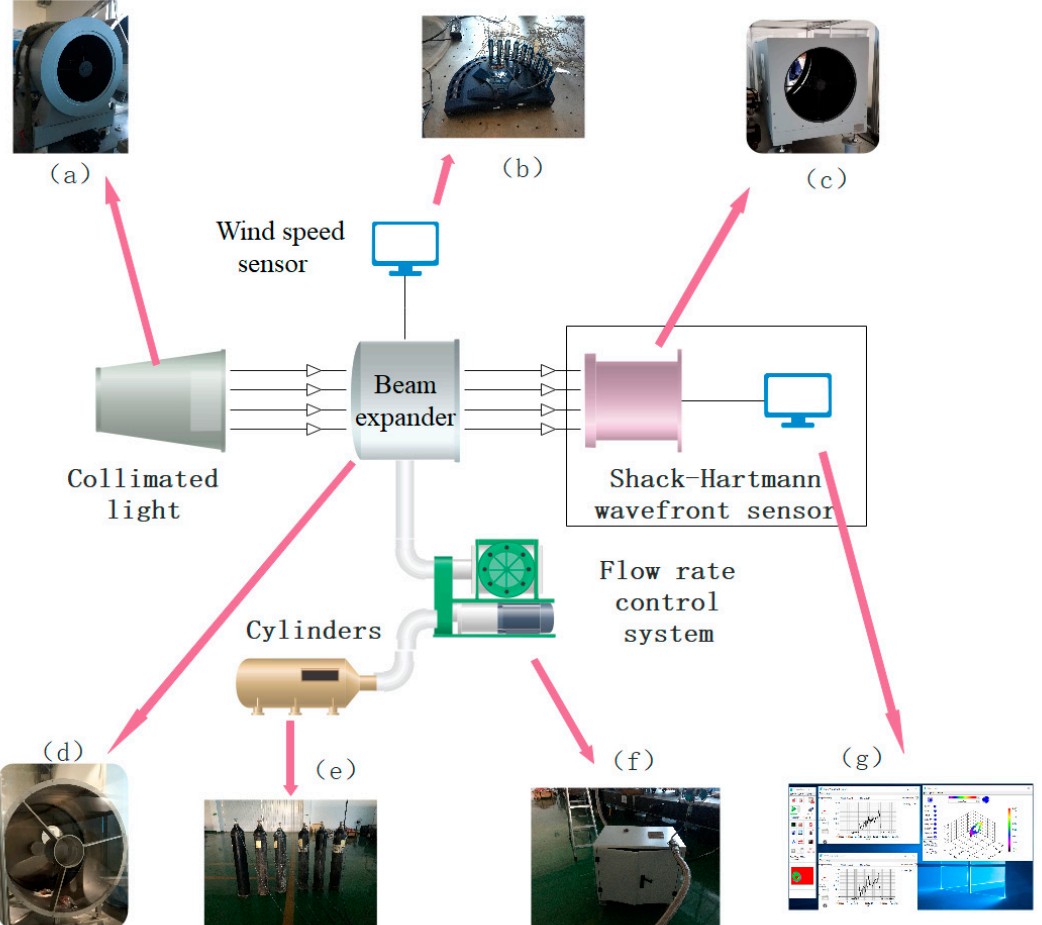

**Figure 9.** Experimental platform and equipment: (**a**) collimator; (**b**) wind speed sensor; (**c**) Shack–Hartmann wavefront sensor; (**d**) beam expander; (**e**) cylinders; (**f**) flow rate control system; (**g**) screen.

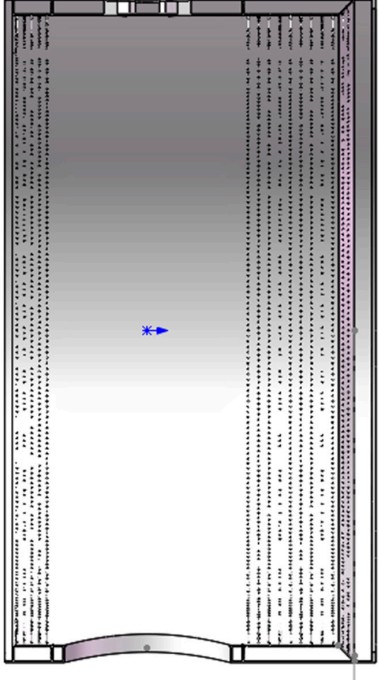

**Figure 10.** Distribution of the intake holes.

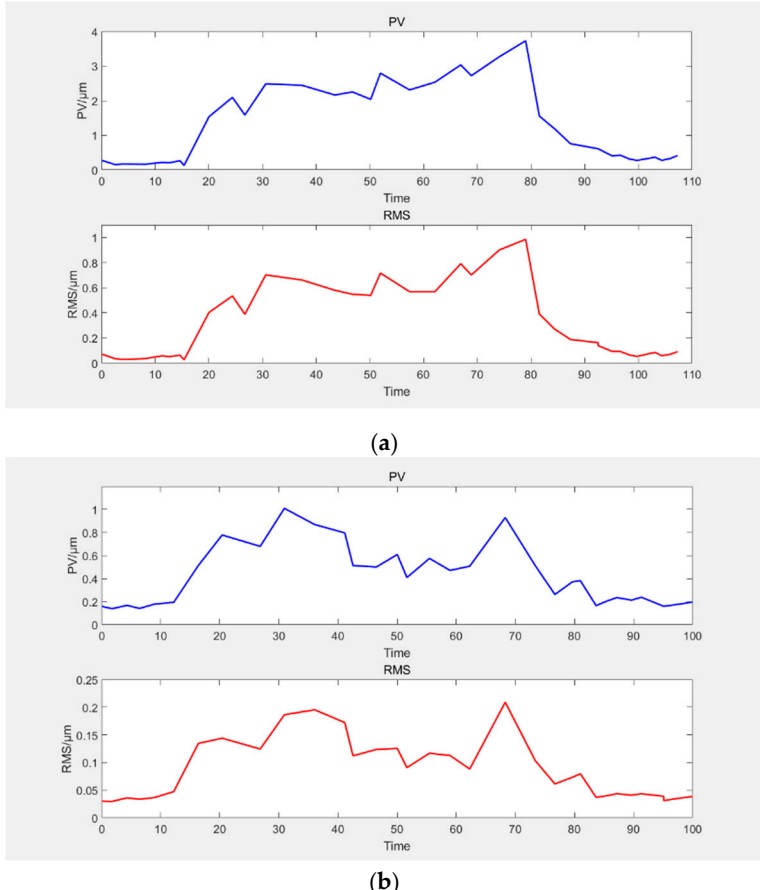

**Figure 11.** Wavefront aberration profiles over time: (**a**) wavefront aberration with tilt; (**b**) wavefront aberration without tilt.

The results of another measurement performed after eliminating the tilt in the experimental set-up are shown in Figure 11b. Because of the processing errors and unevenness of inflation, a larger tilt is introduced into the wavefront aberration. Since the tilt can be eliminated by adjusting the mirror, we no longer consider the tilt in the calculation process. As seen from this figure, the results were consistent with the trend observed in Figure 11a. The wavefront aberration was stable at approximately 0.04 μm before the gas flow was started. When gas intake started at 12 s, the wavefront aberration rose rapidly, and the RMS value finally stabilized at approximately 0.12 μm. When the gas supply was stopped at 62 s, the wavefront aberration initially rose to 0.2 μm and then quickly returned to the pre-flow level.

From the experimental results, we found that the RMS of wavefront aberration caused by the introduced airflow was 0.08 μm while the RMS value estimated from the simulation was only approximately 0.03 μm. Through comparisons of the experimental and simulated scenarios, we found that one important reason for this discrepancy was that the length of ray tracing was insufficient. In our simulation, ray tracing was performed only up to the exit of the beam expanding system, corresponding to a tracing distance of 1151 mm. However, during the experiment, we placed the Shack–Hartmann wavefront sensor one metre from the beam expanding system to prevent the wavefront sensor from affecting the flow field at the outlet of the beam expanding system and to avoid gas expelled from the beam expanding system from blowing into the wavefront sensor. To account for this, we repeated the ray tracing process for plan E with a tracing length of 2151 mm. With this analysis, the RMS of wavefront aberration was found to be 0.077 μm, which is close to the experimental result.

During the experiment, we also used a series of wind speed sensor to monitor the changes in wind speed. The results are shown in Figure 12. When the inflation starts, the airflow speed rises rapidly, and the velocity of the fluid in the system can reach the level of the platform within a short time.

Comparing Figures 12 and 13, it is found that the speed increase at the beginning of the experiment is slower than the simulation. The reason for this phenomenon is that it takes time for the pipeline pressure to increase. Nonetheless, the tendency and the final results of the two are consistent.

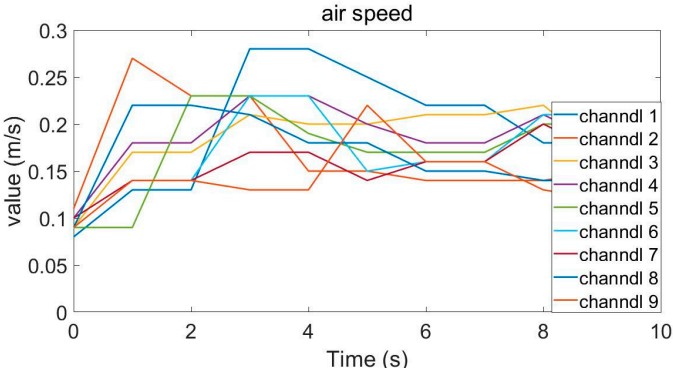

**Figure 12.** Wind speed profiles of experiment.

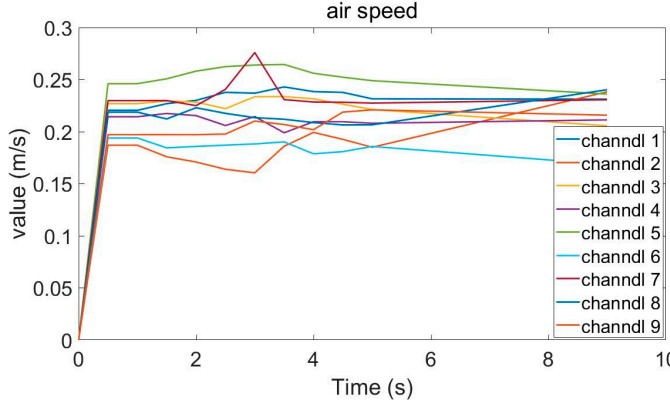

**Figure 13.** Wind speed profiles of simulation.

## 5. Conclusions

In this paper, simulations and experimental studies are reported to verify the influence of the active air intake system on the beam quality in a windowless beam expanding system. Through simulation, we found that the distribution of the inlet holes in the intake system has a great influence on the system's wavefront aberration. We observed that by increasing the number of rows and the number of holes per row for inlet holes, the internal flow field of the beam expanding system can be simplified. Finally, it was found that an air intake layout with 16 rows and 360 holes per row had the optimal effect for the beam expanding system of the 850 mm optical telescope when the mass flow rate was 100 g per second. The RMS of wavefront aberration in this state is only 0.08 μm, which is acceptable in engineering practice.

In this paper, the method of flow field simulation combined with ray tracing is well applied to the simulation of aero-optical effects. The concept of differentiation is applied to divide the ray transmission path after the flow field simulation to avoid fitting the refractive index field with complex formulas. To obtain more accurate results, the Barron gradient operator instead of the surface normal of each element is used to calculate the refractive index gradient. After calculating the direction of the refractive index gradient, Snell's law is directly used to calculate the light deflection to determine the direction of the light emitted from the surface of each element. Compared with other traditional methods, this algorithm is simpler, more accurate, and easier to implement. By comparing the simulation and experimental results, it is obvious that the simulation process can reflect the real situation precisely. Thus, the simulation can be used in future design work for a beam expanding system.

**Author Contributions:** Conceptualization G.R. and Y.T.; methodology, Q.Z.; software, Q.Z.; validation, Y.T., T.T. and G.R.; formal analysis, Q.Z.; investigation, Q.Z.; resources, T.T.; data curation, Q.Z.; writing—original draft preparation, Q.Z.; writing—review and editing, Y.T.; visualization, Q.Z.; supervision, Y.T.; project administration, Y.T.; funding acquisition, Y.T. All authors have read and agreed to the published version of the manuscript.

**Funding:** This research was funded by Open Research Fund of State Key Laboratory of Pulsed Power Laser Technology (SKL2018KF05), Excellent Youth Foundation of Sichuan Scientific Committee (2019JDJQ0012), Youth Innovation Promotion Association, CAS (2018411), CAS "Light of West China" Program and Young Talents of Sichuan Thousand People Program.

**Conflicts of Interest:** The authors declare no conflict of interest.

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
