# Peer review of "Study of the Effect of Intake Layout on the Wavefront in a Beam Expanding System of a Telescope"

_applsci, doi:10.3390/app10103643_

Round 1

Reviewer 1 Report

[1] The abstract shall be re-written. Please refer the guide line of abstract.

[2] The description of numerical simulation domain is insufficient. With figure 2. Schematic description might help for reader’s understanding.

[3] Physical model and boundary conditions shall be introduced which used in this analysis. Gravity was considered? What is the boundary condition of airflow region?

[4] Figure 3-6 are inadequate for corresponding descriptions. Numbers of scale bar is blur. And the scale bar range is too large to comparing different results.

[5] When the simulation was stable? What is the criteria of stable flow? Temporal fluctuation was not seen after stable time?

[6] It seems inappropriate to call optimal condition. Are there any results or restrictions on the increase in rows more than 16?

[7] Comparison with the results of experiments on one condition seems inadequate by means of validation.

[8] What is the tilt condition? No description was given in manuscript.

[9] For the last paragraph of chapter 4, the description of temporal behavior of simulation might be needed.

Reviewer 2 Report

The authors approached the investigation of the air-flow turbulence effect on the wavefront measured on a mid-size telescope
equipped by a standard Cassegrain optical configuration, induced by a beam expanding system when a clear gas injection is
performed. The consistency and correctness of a measured wavefront is crucial to maintain the optical collimation in a perfect
shape and to take optical aberrations under control during observations. That's why the topic focused by the authors is
of extremely importance in the design of high-precision astronomical instruments, especially those having a large aperture.
The proposed solution is consisting into the introduction of a matrix of small holes within the beam expanding system, in order
to reduce the air turbulence effect on the aberrations of the optical path. Different configurations have been well tested,
founding the best solution made by increasing both rows and amount of holes per row of the matrix. This was intuitively
obvious, but authors discussed its justification through a series of accurate tests. Results appear quite interesting,
according to the general discussions in literature regarding the relevance of proper solutions to prevent optical path
distorsions induced by turbulence on the focal plane.
The manuscript is well written, easy to follow and rigorous in terms of test description and results discussion.
We just recommend to enlarge the figures from 3 to 6, in order to make more readable the colour scales.

Author Response

Dear Reviewer:

we want to begin by thanking you for Writing that "The manuscript is well written, easy to follow and rigorous in terms of test description and results discussion." We also appreciated the constructive criticism and suggestion. Weaddressed all the points raised as below:

  1. We just recommend to enlarge the figures from 3 to 6, in order to make more readable the colour scales.

       Response1:according to the referee's suggestion, we have modified the figures from 3 to 6.

Thank you again for your valuable comments.

Round 2

Reviewer 1 Report

The 1st review's comments are well replied and applied in the manuscript. There is no additional comment. Thank you for the authors' contribution to the corresponding research field.